# Antimicrobial Activity of Stilbenes from *Bletilla striata* against *Cutibacterium acnes* and Its Effect on Cell Membrane

**DOI:** 10.3390/microorganisms11122958

**Published:** 2023-12-11

**Authors:** Qian Yu, Luyao Sun, Fu Peng, Chen Sun, Fang Xiong, Meiji Sun, Juan Liu, Cheng Peng, Qinmei Zhou

**Affiliations:** 1State Key Laboratory of Characteristic Chinese Medicine Resources in Southwest China, Chengdu University of Traditional Chinese Medicine, Chengdu 611137, China; 2School of Basic Medicine, Chengdu University of Traditional Chinese Medicine, Chengdu 611137, China; 3Key Laboratory of Drug-Targeting and Drug Delivery System of the Education Ministry and Sichuan Province, Sichuan Engineering Laboratory for Plant-Sourced Drug and Sichuan Research Center for Drug Precision Industrial Technology, West China School of Pharmacy, Sichuan University, Chengdu 610041, China; 4Innovative Institute of Chinese Medicine and Pharmacy, Chengdu University of Traditional Chinese Medicine, Chengdu 611137, China

**Keywords:** *Bletilla striata*, stilbenes, *Cutibacterium acnes*, antibacterial activity, cell membrane

## Abstract

The abnormal proliferation of *Cutibacterium acnes* is the main cause of acne vulgaris. Natural antibacterial plant extracts have gained great interest due to the efficacy and safety of their use in skin care products. *Bletilla striata* is a common externally used traditional Chinese medicine, and several of its isolated stilbenes were reported to exhibit good antibacterial activity. In this study, the antimicrobial activity of stilbenes from *B. striata* (BSS) against *C. acnes* and its potential effect on cell membrane were elucidated by determining the minimum inhibitory concentration (MIC), minimum bactericidal concentration (MBC), bacterial growth curve, adenosine triphosphate (ATP) levels, membrane potential (MP), and the expression of genes related to fatty acid biosynthesis in the cell membrane. In addition, the morphological changes in *C. acnes* by BSS were observed using transmission electron microscopy (TEM). Experimentally, we verified that BSS possessed significant antibacterial activity against *C. acnes*, with an MIC and MBC of 15.62 μg/mL and 62.5 μg/mL, respectively. The growth curve indicated that BSS at 2 MIC, MIC, 1/2 MIC, and 1/4 MIC concentrations inhibited the growth of *C. acnes*. TEM images demonstrated that BSS at an MIC concentration disrupted the morphological structure and cell membrane in *C. acnes*. Furthermore, the BSS at the 2 MIC, MIC, and 1/2 MIC concentrations caused a decrease in the intracellular ATP levels and the depolarization of the cell membrane as well as BSS at an MIC concentration inhibited the expression of fatty acid biosynthesis-associated genes. In conclusion, BSS could exert good antimicrobial activity by interfering with cell membrane in *C. acnes*, which have the potential to be developed as a natural antiacne additive.

## 1. Introduction

Globally, acne vulgaris is an inflammatory skin condition that affects 9.38% of the population, making it the eighth most prevalent disease [1,2]. Acne is mainly caused by the bacterium *Cutibacterium acnes*, a common bacterium found on human skin [3]. It is true that antibiotics are effective against sensitive *C. acnes*, but they are also resistant due to monotherapy and overuse [4]. Among other treatments, retinoids and benzoyl peroxide are limited by patient compliance because of unpleasant side effects [5]. As a result, there is a need for new effective treatments that have fewer side effects.

*Bletilla striata*, a perennial herb of the genus *Bletilla* (Orchidaceae), is mainly distributed in China, including the Guangxi, Yunnan, and Sichuan provinces [6]. Its dry tubers have been commonly used in traditional medicine for centuries to treat various diseases, including lung disorders, gastrointestinal disorders, traumatic bleeding, burns, and ulcers [7,8]. It has been reported that *B. striata* contains stilbenes (bibenzyls and phenanthrenes) and polysaccharides [9,10]. There has been evidence that stilbenes from *B. striata* (BSS) possess anti-inflammatory, antitumor, and antibacterial properties [11,12], indicating that BSS has medical benefits to be promising healthcare products.

But there have been few studies evaluating the effectiveness of BSS for *C. acnes*. Therefore, the main active components of BSS were identified and analyzed by HPLC/Q-TOF-MS in the study. The inhibitory effect of BSS on *C. acnes* was tested in vitro, and its potential effect on cell membrane was further explored. This study provides new insights into how BSS may act against *C. acnes*.

## 2. Materials and Methods

### 2.1. Chemicals

Doxycycline hydrochloride (purity ≥ 98.0%, Lot No: D14332) was purchased from Frontier Scientific, Inc. (Newark, DE, USA). 2,3,5-Triphenyltetrazolium chloride (TTC, purity ≥ 98.0%, Lot No: BCCH3368) was acquired from Sigma-Aldrich (Wien, Austria).

### 2.2. Bacterial Strains and Culture

*C. acnes* (ATCC 11827) was obtained from Guangdong Microbial Culture Collection Center (Guangzhou, China). It was anaerobically incubated in a Brain Heart Infusion (BHI) Broth (Beijing Solarbio Science & Technology Co., Ltd., Beijing, China) agar plate at 37 °C.

### 2.3. Extraction of Stilbenes from B. striata

*B. striata* was procured from the Sichuan Provincial Chinese Medicine Drinking Tablets Co., Ltd. and identified by the Associate Professor Gao Jihai of Chengdu University of Traditional Chinese Medicine. The tubers of *B. striata* (5 kg) were extracted with 65% ethanol three times under reflux, for 1 h each time. The extract was evaporated under reduced pressure to yield a residue that was later suspended in water. The solution was then loaded to a polyamide chromatography column. Afterwards, the column was eluted with 20% ethanol to remove the impurities. And, the mobile phase of 95% ethanol was collected until the stilbenes were completely eluted. The solution was filtered and concentrated to obtain 20 g of extract with a content of 0.4%.

### 2.4. Instrumentation and HPLC/Q-TOF-MS Conditions

High-performance liquid chromatography (HPLC) was performed using an Agilent 1260-Bruker tims-TOF-MS system, equipped with a binary solvent delivery system, an autosampler, and a DAD system. The separation was conducted on an Agilent Poroshell 120 EC-C18 column (4.6 mm × 150 mm, 4 μm, Agilent, Santa Clara, CA, USA). The mobile phase consisted of (A) methanol and (B) 0.1% glacial acetic acid in water. The HPLC elution conditions were as follows: 30–95% A (0–55 min) and 95–100% A (55–60 min). The flow rate was kept at 0.4 mL/min. The column was maintained at 40 °C. Mass spectra were obtained using a high-definition mass spectrometer (HDMS) (Bruker Daltonics, Billerica, MA, USA) equipped with an electrospray ionization source. Electrospray ionization (ESI) mass spectra were acquired over the *m*/*z* 50–1300 range. Negative mode was obtained and the capillary voltage was set to 3000 V.

### 2.5. Determination of MIC and MBC

The MIC of BSS against *C. acnes* was determined by the broth microdilution method combined with the triphenyl tetrazolium chloride (TTC) indicator method [13,14]. The serial two-fold dilutions of BSS were prepared in 96-well plates (100 μL per well) using BHI broth as a diluent, followed by adding 100 μL of bacterial suspension (1.5 × 10^6^ CFU/mL) to each well to obtain the final concentrations of 125.00–0.49 μg/mL. Doxycycline hydrochloride (2.50–0.01 μg/mL) was used as the positive control and 0.2% DMSO was used as the negative control. After anaerobic incubation at 37 °C for 48 h, 30 μL 0.5% (*w*/*v*) TTC solution was added to the wells as an indicator and cultured at 37 °C for 1.5 h. Tetrazolium salts can be reduced by active bacteria to red-colored formazan, so the MIC is calculated as the lowest drug concentration in the wells that did not turn red. Afterwards, MBC was determined by inoculating 10 μL culture on BHI agar plates and incubating it anaerobically at 37 °C for 48–72 h. The MBC is considered to be the lowest drug concentration at which no bacteria grew.

### 2.6. Determination of Growth Curve

The effect of BSS on the growth of *C. acnes* was assessed by determining the growth curve [15]. One hundred microliters of BSS at different concentrations were added to 96-well plates, followed by adding 100 μL of bacterial suspension (1.5 × 10^6^ CFU/mL) to each well to obtain the final concentrations of 2 MIC, MIC, 1/2 MIC, and 1/4 MIC, and a bacterial suspension containing 0.2% DMSO was used as the control. The cultures were anaerobically incubated at 37 °C for 72 h, and the absorbance values were measured at 600 nm every 4 h during the incubation period using the FlexStation 3 multimode microplate reader (Molecular Devices, San Jose, CA, USA).

### 2.7. Observation of TEM

The effect of BSS on the morphology of *C. acnes* was observed by TEM [16]. Bacterial suspension (1.5 × 10^8^ CFU/mL) was treated with the MIC concentration of BSS, and bacterial suspension containing 0.2% DMSO was used as the control. After anaerobic incubation at 37 °C for 48 h, the bacteria were collected by centrifugation (6000 rpm, 4 °C, 10 min) and washed with PBS (0.01 M, pH 7.2), fixed overnight with 2.5% glutaraldehyde, refixed with 1% osmium tetroxide. This was followed by progressive dehydration using acetone solution, and then permeabilized, embedded, sectioned, and stained. Finally, the samples were observed using the JEM-1400FLASH TEM (JEOL, Akishima, Japan).

### 2.8. Determination of Intracellular ATP Levels of C. acnes

The effect of BSS on the intracellular ATP levels of *C. acnes* was determined by BacTiter-Glo™ Microbial Cell Viability Assay (Promega Corporation, Madison, WI, USA) [17]. One hundred microliters of bacterial suspension (1.5 × 10^8^ CFU/mL) containing 2 MIC, MIC, and 1/2 MIC concentrations were pipetted into 96-well plates, respectively. And, bacterial suspension containing 0.2% DMSO was used as the control. After anaerobic incubation at 37 °C for 3 h, 100 µL of BacTiter-Glo™ reagent was added. The sample’s ATP luminescence was measured using the FlexStation 3 multimode microplate reader and expressed in relative luminescence units (RLUs).

### 2.9. Measurement of MP

The effect of BSS on MP of *C. acnes* was analyzed by measuring Rhodamine fluorescence value [18]. Bacterial suspension (1.5 × 10^8^ CFU/mL) was treated with BSS at the final concentrations of 2 MIC, MIC, and 1/2 MIC, and bacterial suspension containing 0.2% DMSO was used as the control. After anaerobic incubation at 37 °C for 3 h, bacterial suspension was centrifuged (6000 rpm, 5 min), washed, and resuspended with PBS. Rhodamine 123 (Shanghai Yuanye Bio-Technology Co., Ltd., Shanghai, China) was added to the final concentration of 2 μg/mL. After incubation for another 30 min under dark conditions, the samples were washed and resuspended with PBS. Subsequently, the sample’s fluorescence intensity was measured at excitation and emission wavelengths of 480 nm and 530 nm using the FlexStation 3 multimode microplate reader.

### 2.10. Real-Time Quantitative PCR (qRT-PCR) Analysis

A qRT-PCR analysis of *C. acnes* was conducted to determine whether BSS affected the expression of fatty acid biosynthesis genes [19]. *C. acnes* was incubated to the logarithmic growth phase before being treated with BSS at a concentration of MIC for 12 h, and the bacterial suspension containing 0.2% DMSO was used as the control. According to the manufacturer’s instructions, the total RNA of *C. acnes* was extracted with Trizol reagent (Beyotime Biotechnology, Shanghai, China) and its concentration was measured; then, RNA was reverse transcribed into cDNA using the High Capacity cDNA Reverse Transcription Kit (Thermo Fisher Scientific, Vilnius, Lithuania). Finally, the expression of fatty acid biosynthesis-related genes (*acpS*, *fabD*, *fabH*, *fabG,* and *fabI*) was determined by qRT-PCR using PowerUp^TM^ SYBR^TM^ Green Master Mix (Thermo Fisher Scientific, Vilnius, Lithuania). *gyrB* was used as the housekeeping gene. Gene expression was analyzed by 2^−ΔΔCt^ method. Primers are shown in Table 1.

### 2.11. Statistical Analysis

All experiments were performed in triplicate and data are expressed as the mean ± SD. Statistical analysis was performed using SPSS 22.0 software, and differences were considered statistically significant at *p* < 0.05 or *p* < 0.01.

## 3. Results

### 3.1. HPLC/Q-TOF-MS Profile of BSS

The HPLC chromatogram of BSS revealed multiple peaks with a retention time from 0 to 60 min (Figure 1). According to the MS spectra and retention time, ten components were identified in stilbenes, including Coelonin, 2,4,7-trimethoxy-9,10-dihydrophenanthrene, Bulbocodin D, etc. The identification and chemical components in the stilbenes sample are shown in Table 2.

### 3.2. Antimicrobial Activity of BSS against C. acnes

The antibacterial activity of BSS against *C. acnes* was evaluated by determining MIC and MBC. According to the experimental results, doxycycline hydrochloride had an MIC of 0.31 μg/mL and an MBC of 0.63 μg/mL against *C. acnes*, and those of BSS were 15.62 μg/mL and 62.5 μg/mL, respectively, demonstrating that BSS had a favorable bacteriostatic activity against *C. acnes*.

### 3.3. Inhibitory Effect of BSS on the Growth of C. acnes

The growth curve of *C. acnes* was plotted to observe the inhibitory effect of BSS on *C. acnes* growth. As shown in Figure 2, the growth of *C. acnes* in the control group approximately followed the model S-shaped growth curve, reaching the logarithmic growth phase after 16 h and the stable phase after 64 h. Compared with the control group, the growth of *C. acnes* was inhibited to different degrees after the treatment with 2 MIC, MIC, 1/2 MIC, and 1/4 MIC concentrations of BSS. Notably, BSS at 2 MIC and MIC concentrations almost inhibited the growth of *C. acnes*. The experimental results indicated that BSS was effective in suppressing the growth of *C. acnes*.

### 3.4. Effect of BSS on the Morphology of C. acnes

The TEM technique was used in this study for the direct observation of changes in the morphological structure of *C. acnes* by BSS. As shown in Figure 3, the morphological structure of *C. acnes* in the control group was normal and rod-shaped, with a complete and smooth cell surface and abundant cytoplasmic contents. In comparison with the control group, the BSS treatments of *C. acnes* led to deformed, curled, and ruptured cells as well as the loss of cytoplasmic contents. The experimental results revealed that BSS could affect the bacterial growth and survival by altering its normal morphological structure as well as destroying the cell wall and cell membrane.

### 3.5. Effect of BSS on the Intracellular ATP Levels of C. acnes

ATP is the main source of bioenergy for various bacteria. The ATP levels reflect bacterial viability and growth and decrease when the bacteria are damaged or dying [20]. The BacTiter-Glo™ Microbial Cell Viability Assay is a luciferase-based assay to quantify ATP levels [21]. In this study, the changes in the BSS on the intracellular ATP levels of *C. acnes* were measured by ATP luminescence (RLU). As shown in Figure 4, compared with the control group, the intracellular ATP levels of *C. acnes* were remarkably reduced in a concentration-dependent manner after the treatment with BSS (*p* < 0.01). The experimental results suggested that BSS could interfere with bacterial growth and cell viability by decreasing intracellular ATP levels.

### 3.6. Effect of BSS on MP of C. acnes

Rhodamine 123 is a lipophilic cationic dye, and changes in its fluorescence intensity reflect changes in bacterial MP [22]. A decrease in MP indicates membrane depolarization; conversely, an increase in MP indicates membrane hyperpolarization [23]. Consequently, Rhodamine 123 was used in this study to investigate the effect of BSS on the MP of *C. acnes*. As expected, the MP of *C. acnes* treated with BSS was reduced in a concentration-dependent manner (Figure 5). Compared to the control group, the mean fluorescence intensity (MFI) of the 2 MIC and MIC groups was reduced by 37.61% and 21.27% (*p* < 0.05 or *p* < 0.01), respectively. The experimental results suggested that BSS could arouse the marked depolarization of the cytoplasmic membrane in *C. acnes*, leading to cell membrane damage.

### 3.7. Effect of BSS on Fatty Acid Biosynthesis-Related Gene Expression of C. acnes

Fatty acid synthesis (FAS) in bacteria is vital for cell membrane production [24]. In order to further explore the effect of BSS on cell membrane in *C. acnes*, the fatty acid biosynthesis pathway after the exposure to BSS was analyzed. As shown in Figure 6, compared with the control group, BSS significantly downregulated the expression of *acpS* (encoding holo-ACP synthase), *fabD* (encoding malonyl-CoA-ACP transacylase), *fabH* (encoding β-ketoacyl-ACP synthase III), *fabG* (encoding 3-oxoacyl-ACP reductase), and *fabI* (encoding enoyl-ACP reductase) (*p* < 0.01). The experimental results indicated that BSS could interfere with fatty acid biosynthesis in the cell membrane by downregulating the gene expression of *acpS*, *fabD*, *fabH*, *fabG*, and *fabI*.

## 4. Discussion

Acne is a common skin disease which influences the quality of life and psychosocial function of patients. Usually, chemical drugs can only inhibit microbial infections, suppress the inflammatory response, or reduce the accompanying symptoms, but overuse and tolerance are also problems that cannot be ignored. Therefore, plant extracts have gained popularity because of the greater efficacy and safety of their use in skin care products. Stilbenes are secondary metabolites produced by plants in response to external stimuli such as fungal infection, trauma, and so on [25,26]. Notably, there is growing evidence that stilbenes are abundant in *B. striata*, which received much attention because of their good biological activities [27]. Modern pharmacological research reveals that BSS has a good antibacterial effect against a variety of bacteria such as methicillin-resistant *Staphylococcus aureus*, *S. aureus*, *Bacillus subtilis*, *Escherichia coli*, *Bacillus cereus* var. *mycoides, Nocardia gardneri,* and so on [28,29,30]. However, their suppressive effect on *C. acnes* has not been investigated to date. In present study, the MIC and MBC exhibited that BSS had a favorable antimicrobial activity, with MIC and MBC of 15.62 μg/mL and 62.50 μg/mL, respectively. The growth curve showed that BSS had a significant growth inhibitory effect on *C. acnes*.

Since BSS has good antibacterial activity against *C. acnes*, we further analyzed the effects of BSS on the bacterial cell membrane. Using TEM, we observed significant morphological changes in *C. acnes* when exposed to BSS at MIC concentration, with the severe bending and deformation of the bacteria, the rupture of the cell membrane, and the leakage of cytoplasmic contents. As we know, the cell membrane, as a barrier to bacterial life activities, plays a crucial role in maintaining bacterial material and energy homeostasis and is a key target of many antimicrobial agents [31]. MP is the potential difference between the interior and exterior of a biological cell membrane and is significant for ATP production and cellular functioning [32,33]. It has been reported that a change in the external environment of bacteria alters the ion balance on both sides of the membrane, resulting in an alteration of MP [34]. Igbinosa et al. showed that *Streptomyces* extracts induced cell membrane depolarization, which resulted in cell membrane disruption and bacterial death [35]. In this present study, we found that treatment with BSS caused the depolarization of the cell membrane, which further damaged the cell membrane and affected bacterial survival.

ATP, as a major energy carrier in organisms, is essential for bacterial metabolism and physiological activities [36]. In *C. acnes*, the anaerobic electron transport chain generates ATP from the Wood–Werkman cycle, and electrons are transferred from NADH to intracellular fumarate via NADH dehydrogenase and fumarate reductase to drive the proton pump for ATP synthesis [37]. It has been suggested that the decrease in intracellular ATP levels may be attributed to a slower rate of ATP synthesis and a faster rate of ATP hydrolysis, as well as cell membrane damage [38]. In our study, it was clearly demonstrated that treatment with BSS induced a decrease in the intracellular ATP levels in *C. acnes*, leading to a reduction in cell viability as well as affecting bacterial survival.

Fatty acid is an important ingredient for bacteria to synthesize various phospholipids in the cell membrane, and some bacteriostatic agents can target and inhibit the key enzymes in the fatty acid synthesis (FASII) pathway to inhibit fatty acid biosynthesis and achieve the bacteriostatic effect [39,40]. Therefore, scholars at home and abroad have taken the FASII pathway as the focus of developing bacteriostatic substances. The FASII pathway consists of a series of enzymes that sequentially and cyclically recognize and catalyze fatty acid carbon chain substrates covalently carried by acyl carrier protein (ACP) to achieve the synthesis and extension of saturated or unsaturated fatty acid carbon chains of a specific length [41,42]. According to the synthesis process in FASII, FASII is mainly divided into two phases: initiation and elongation cycle. *acpS*, *fabD*, and *fabH* are pivotal genes for the initiation phase of FASII. *acpS* encodes holo-ACP synthase, which generates holo-ACP by transferring the 4′-phosphopantetheine group of coenzyme A (CoA) to the serine residue of apo-ACP [43]. After apo-ACP is converted to holo-ACP, *fabD* encodes malonyl-CoA-ACP transacylase, which transfers the malonyl group of malonyl-CoA to holo-ACP to form malonyl-ACP [44]. *fabH* encodes β-ketoacyl-ACP synthase III, which catalyzes the condensation of malonyl-ACP and acetyl-CoA and is a key enzyme linking the initiation phase and the elongation cycle phase of FASII [45]. *fabG* and *fabI* are pivotal genes for the elongation cycle phase of FASII. *fabG* encodes 3-oxoacyl-ACP reductase, which reduces β-ketoacyl-ACP to β-hydroxyacyl-ACP [46]. *fabI* encodes enoyl-ACP reductase, which reduces enoyl-ACP to saturated acyl-ACP and is the final step in the elongation cycle phase of FASII [47]. In recent years, the FASII pathway has attracted increasing attention as an important direction for the development of antimicrobial agents. It was reported that anthranilic acid derivatives targeting *acpS* hindered CoA binding and catalyzing reactions by competitively binding to the CoA binding site [48]. Pang et al. found that Sanggenon D could suppress the initiation of fatty acid biosynthesis by downregulating *fabD* and *fabH* expression, as well as by regulating fatty acid carbon chain elongation by downregulating *fabG* and *fabI*, which further disrupted the cell membrane of *S. aureus* [49]. In this present study, the qRT-PCR results revealed that BSS could break the cell membrane in *C. acnes* by altering the expression of genes involved in fatty acid biosynthesis.

Although chemical drugs are useful for treatments seeking to prevent acne, the widespread use of antibiotics has led to drug resistance. BSS has been shown to exhibit significant antibacterial effects against *C. acnes*, which could emerge as new products for acne treatment. As a natural extract, BSS may have a higher efficacy and a lower toxicity for replacing or assisting antibacterial agents. However, further investigations need to be carried out to clarify their application value.

## 5. Conclusions

This study provided evidence that BSS contributed to suppressing the proliferation of *C. acnes*. Notably, BSS impacted the cell membrane of *C. acnes* by altering MP and modulating the gene expression of fatty acid biosynthesis. In addition, TEM images demonstrated that BSS disrupted the cell membrane in *C. acnes*. Taken together, stilbenes could be a natural antibacterial agent to control the negative impact of *C. acnes* in medicine and healthcare.

## Figures and Tables

**Figure 1 microorganisms-11-02958-f001:**
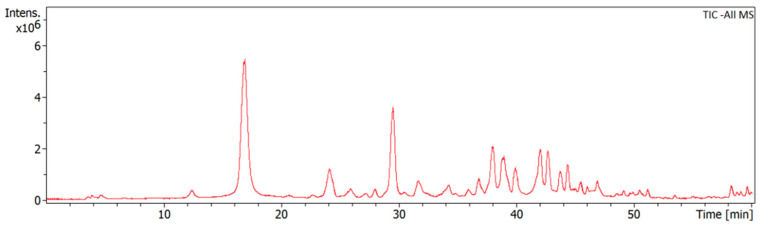
HPLC/Q-TOF-MS analysis of BSS.

**Figure 2 microorganisms-11-02958-f002:**
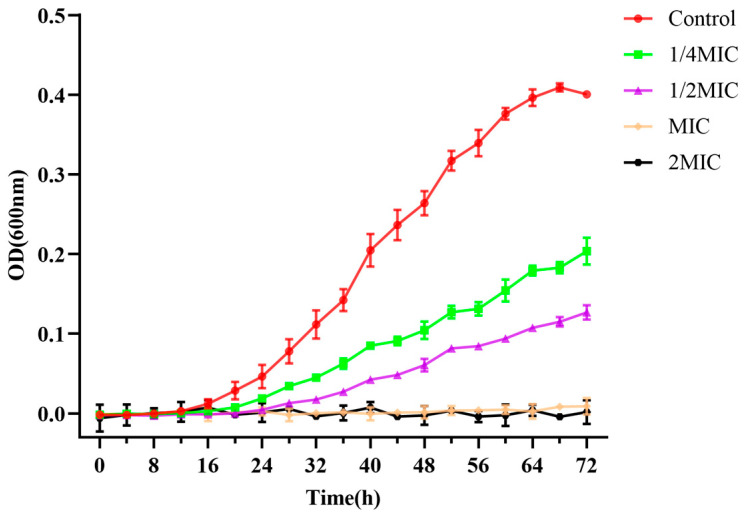
Growth curve of *C. acnes* treated with different concentrations of BSS. Bars represent the standard deviation (*n* = 3).

**Figure 3 microorganisms-11-02958-f003:**
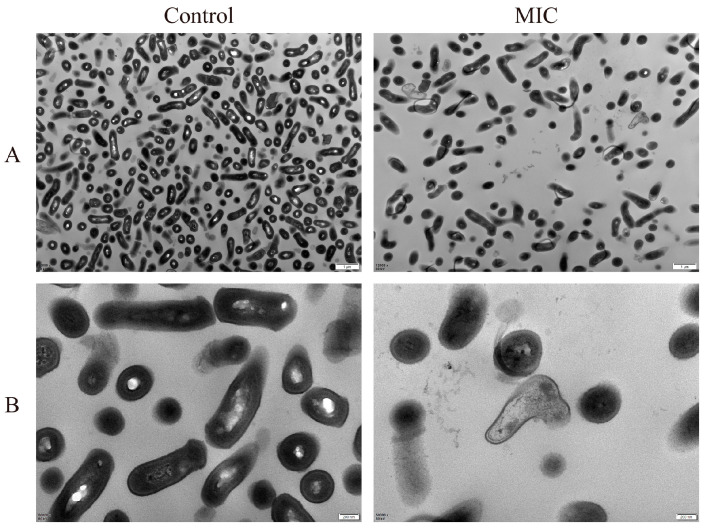
Effect of BSS on the morphology of *C. acnes*. (**A**) TEM image of *C. acnes* at 12,000× magnification; scale bar: 1 μm. (**B**) TEM image of *C. acnes* at 50,000× magnification, scale bar: 200 nm.

**Figure 4 microorganisms-11-02958-f004:**
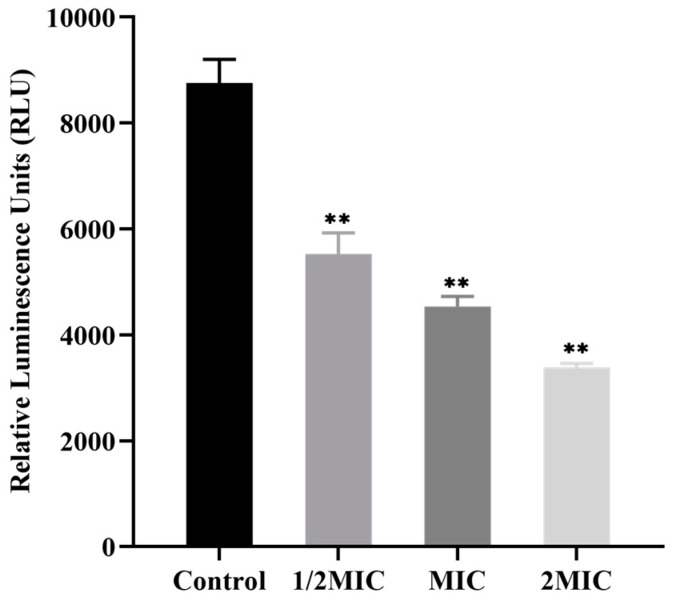
Effect of BSS on intracellular ATP levels of *C. acnes*. Data are expressed as the mean ± SD (*n* = 3). ** *p* < 0.01 indicates significant differences compared with the control group.

**Figure 5 microorganisms-11-02958-f005:**
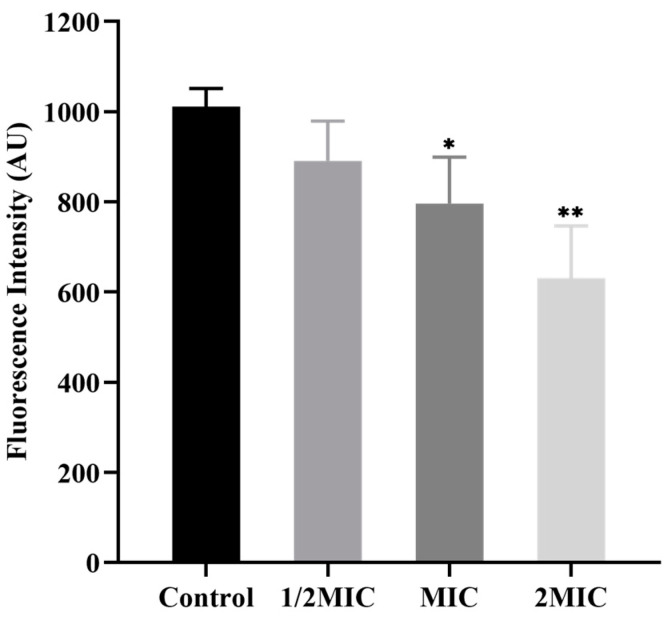
Effect of BSS on the MP of *C. acnes*. Data are expressed as the mean ± SD (*n* = 3). * *p* < 0.05 and ** *p* < 0.01 indicate significant differences compared with the control group.

**Figure 6 microorganisms-11-02958-f006:**
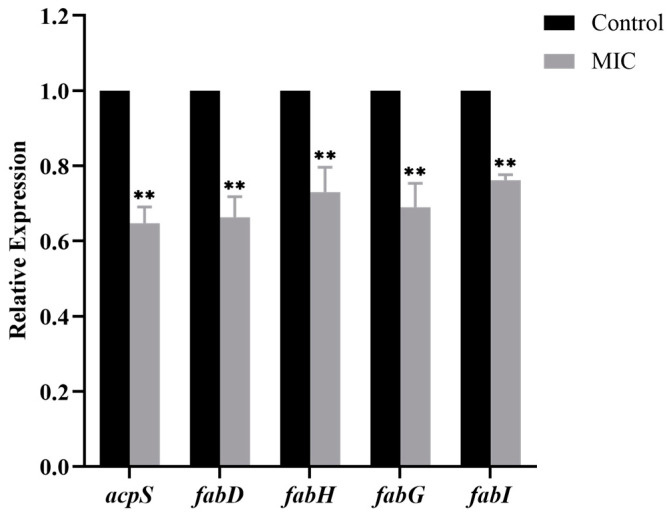
Effect of BSS on fatty acid biosynthesis-related gene expression of *C. acnes*. Data are expressed as the mean ± SD (*n* = 3). ** *p* < 0.01 indicates significant differences compared with the control group.

**Table 1 microorganisms-11-02958-t001:** Primers used in qRT-PCR.

Gene	Forward Primer Sequence (5′-3′)	Reverse Primer Sequence (5′-3′)
*gyrB*	CTGCCCCAAGTTAACCCGAT	TGGCATTGCGCTGAATGAAC
*acpS*	TTGGGTGGAGGAGAAGGACT	ACCTTCGCAGAACACCATCG
*fabD*	CAGCCAACCACAATGGCAAA	CGGTTCCATATGAGCCGTGT
*fabH*	TTGCTACTCAACCGCTCTGG	CAGAGAAGAGGAAGGCGGTG
*fabG*	GTCAGTGGTAGGTCTGCTCG	AAACAAGCTTCTCCGGCAGT
*fabI*	CGATGCACACCTCTATCGCT	GGATCGAGTTTGCGAACAGC

**Table 2 microorganisms-11-02958-t002:** Chemical composition of major stilbenes in BSS.

No	Component Name	Retention Time (min)	Molecular Formula	[M–H]^−^
1	Coelonin	29.5	C_15_H_14_O_3_	241.0697
2	2,4,7-Trimethoxy-9,10-dihydrophenanthrene	34.2	C_17_H_18_O_3_	269.0623
3	1,3-Dimethoxy-5-phenethylbenzene	35.9	C_16_H_18_O_2_	241.0693
4	3,5-Dimethoxybibenzyl	38.0	C_15_H_16_O_3_	243.0849
5	3,3′-Dihydroxy-4-(p-hydroxybenzyl)-5-methoxybibenzyl	42.0	C_22_H_22_O_4_	349.1187
6	3,3′-Dihydroxy-2-(*p*-hydroxybenzyl)-5-methoxybibenzyl	42.7	C_22_H_22_O_4_	349.1185
7	Bulbocodin D	43.7	C_29_H_28_O_5_	455.1524
8	3′,5-Dihydroxy-2-(*p*-hydroxybenzyl)-3-methoxybibenzyl	44.7	C_22_H_22_O_4_	349.1186
9	Gymconopin D	49.1	C_23_H_24_O_4_	363.1329
10	Bulbocol	49.9	C_23_H_24_O_4_	363.1329

## Data Availability

All data included in this study are available upon request through contact with the corresponding author.

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
