# Peer review of "Antimicrobial Activity of Stilbenes from Bletilla striata against Cutibacterium acnes and Its Effect on Cell Membrane"

_microorganisms, 2023, doi:10.3390/microorganisms11122958_

Round 1

Reviewer 1 Report

Comments and Suggestions for Authors

I have carefully reviewed the manuscript entitled “Antimicrobial Activity and Mechanism of Stilbenes from Bletilla striata against Cutibacterium acnes” which focuses on the the antibacterial activity and mechanism of stilbenes from B. striata (BSS) against C. acnes. Moreover, the authors need to response the following comments before consider this Journal.

Comments

1.      In abstract section must be provided quantitative information.

2.      Scale bar should be mentioned clearly in Figure 3.

3.      Why are natural plant extracts gaining popularity in skin care products?

4.      What concentrations of BSS were found to have significant inhibitory effects on C. acnes, and how were they determined?

5.      How does BSS affect the cell membrane of C. acnes, and what role does the membrane play in bacterial life activities?

6.      The author should describe the FASII pathway and its importance in bacterial growth and survival.

7.      Discuss the potential advantages and disadvantages of using BSS compared to chemical drugs for acne treatment based on the information provided.

8.      The conclusion should be revised with outstanding point of this work.

9.      Typographical errors and superfluous spaces throughout the manuscript should be corrected.

Author Response

Dear Reviewer,

Thank you very much for taking the time to review this manuscript entitled “Antimicrobial Activity and Mechanism of Stilbenes from Bletilla striata against Cutibacterium acnes”. We have revised the manuscript based on your comments and made the corresponding responses according to your comments point by point in the following. All the changes have been revised in the manuscript.

Comments 1: In abstract section must be provided quantitative information.

Response 1: Thank you for pointing this out. We agree with this comment. Therefore, we have added BSS-related concentration information to in abstract section, which can be found on page 1, lines 26-30 of the revised manuscript.

Comments 2: Scale bar should be mentioned clearly in Figure 3.

Response 2: Thank you for your good suggestion.We have added scale bar information, which can be found on page 6, line 197 of the revised manuscript.

Comments 3: Why are natural plant extracts gaining popularity in skin care products?

Response 3: Thank you for good comment. Botanical ingredients are one of the main sources of materials that are used in the cosmetics and pharmaceutical industries. Plant extracts can be applied topically for skin care purposes, as well as for the treatment of numerous skin diseases. Their advantage is that they are gentle but effective, safe and non-toxic, without side effects. Cosmetics fortified with bioactive compounds are ideally suited to the needs of the skin and are more environmentally friendly than conventional cosmetics. In addition, plant extracts may exhibit a wide range of properties, both medicinal (in certain skin disorders, including inflammatory disorders such as acne, psoriasis or atopic dermatitis) and for use in skin care (e.g., antioxidant, antibacterial, astringent, moisturizing, regenerating, cleansing, smoothing or lightening). As a result, there is a growing interest in skin care products made from plant extracts.

Comments 4: What concentrations of BSS were found to have significant inhibitory effects on C. acnes, and how were they determined?

Response 4: Thank you for your kind and helpful comment. In this study, the results of growth curve measurements showed that BSS showed almost no growth of bacteria at MIC and 2MIC concentrations, in addition, bacteria at 1/2MIC and 1/4MIC concentrations grew more slowly compared to the control group, and these results responded to the significant inhibitory effect of BSS on bacteria. Furthermore, we have modified the corresponding expression in the manuscript, which can be found on page 5, lines 180-181 of the revised manuscript.

Comments 5: How does BSS affect the cell membrane of C. acnes, and what role does the membrane play in bacterial life activities?

Response 5: Thank you for your good comment. Answer to question 1: In this study, BSS impacted the cell membrane of C. acnes by altering MP and modulating the gene expression of fatty acid biosynthesis. In addition, TEM images showed that BSS ruptured the cell membrane of C. acnes. Answer to question 2: Bacterial cell membrane acts as an osmotic barrier for the cell and operates as a transmitter of soluble substances. It regulates the transmission of the cellular products against the extracellular environment. Bacterial cell membrane is also the site of electron transferring chain and production of ATP or the site of cell proliferation and division.

Comments 6: The author should describe the FASII pathway and its importance in bacterial growth and survival.

Response 6: Thank you for pointing this out. We agree with this comment. Therefore, we have described in this paper the importance of the FASII pathway for C. acnes, which can be found on page 9, lines 279-283 of the revised manuscript.

Comments 7: Discuss the potential advantages and disadvantages of using BSS compared to chemical drugs for acne treatment based on the information provided.

Response 7: Thanks for your valuable comments which have greatly helped us improve this manuscript. We have discussed the potential advantages and disadvantages of using BSS compared to chemical drugs for acne treatment on page 9, lines 309-314 of the revised manuscript.

Comments 8: The conclusion should be revised with outstanding point of this work.

Response 8: Sincerely thank you for your helpful suggestion.We have revised the concluding part of the manuscript. The following changes were made in the revised manuscript: lines 31-32 on page 1, lines 206-207 on page 6, lines 232-234 on page 8, and lines 316-319 on pages 9-10.

Comments 9: Typographical errors and superfluous spaces throughout the manuscript should be corrected.

Response 9: Thank you for your careful review which was helpful in refining the manuscript. We have reviewed the manuscript and removed extra spaces and blank lines.

We hope you'll accept these revisions. If you have any questions, please feel free to contact us. Thank you for your kind consideration of our manuscript.

Reviewer 2 Report

Comments and Suggestions for Authors

Dear Editor,

The results don't support the antibacterial activity of the selected compound. But all the experiments are well controlled. The study authors should change the conclusion of their experiments. 

Future studies with other bacterial strains will help test the antibacterial activity of this compound.

Thank you for the opportunity to review this interesting manuscript. Please let me know if you require any additional information.

Best wishes,

Author Response

Dear Reviewer,

Thank you very much for taking the time to review this manuscript entitled “Antimicrobial Activity and Mechanism of Stilbenes from Bletilla striata against Cutibacterium acnes”. We have revised the manuscript based on your comments and made the corresponding responses according to your comments in the following. All the changes have been revised in the manuscript.

Comments 1: The study authors should change the conclusion of their experiments.

Response 1: Thank you for pointing this out. We agree with this comment. Therefore, We have modified the experimental conclusions of this paper. The following changes were made in the revised manuscript: lines 31-32 on page 1, lines 206-207 on page 6, lines 232-234 on page 8, and lines 316-319 on pages 9-10.

We hope you'll accept these revisions. If you have any questions, please feel free to contact us. Thank you for your kind consideration of our manuscript.

Reviewer 3 Report

Comments and Suggestions for Authors

Dear Authors

I have read your manuscript Antimicrobial Activity and Mechanism of Stilbenes from Bletilla striata against Cutibacterium acnes.

These are my suggestions:

Abstract

Line 16: Change to "owing to".

Line 17: Change to "common".

Line 18: I think you mean "antibacterial activity".

Lines 21-22: Isn't it the C. acnes bacterium that changes morphologically rather than BSS? Please reformulate.

Lines 26 and 28: Change "BBS" to "BSS" if applicable.

Introduction

Lines 45 to 46: Change sentence to "Its dry tubers have been commonly used in traditional medicine for centuries to treat various diseases, including lung disorders, gastrointestinal disorders, traumatic bleeding, burns, and ulcers."

Line 49: Omit either "There's evidence that" or "are known to".

Line 51: Change to "BSS has medical benefits that may be used for promising healthcare products".

Line 54: Change to "The inhibitory effect ..."

Lines 55 to 56: Change sentence to "This study provides new insights into how BSS may act against C. acnes."

Materials and methods

Line 73: Change to "polyamide chromatography column ".

Line 105: Spell out numbers at the beginning of sentences (in this case change to "One hundred microlitres".

Lines 118-119: Change to "This was followed by progressive dehydration using acetone solution", if applicable.

Discussion

Lines 241-242: Change "extracts from natural plants" to "plant extracts".

Lines 245-246: Change "for" to "because of".

Lines 246-247: Since there is more than one BSS, use the plural tense.

Lines 255-246: Change to "Using TEM, we observed significant morphological changes in C. acnes when exposed to BSS at MIC concentration ...".

Line 264: Change to "In this present study ..."

Comments on the Quality of English Language

I have suggested some changes to the English language used in the manuscript.

Author Response

Dear Reviewer,

Thank you very much for taking the time to review this manuscript entitled “Antimicrobial Activity and Mechanism of Stilbenes from Bletilla striata against Cutibacterium acnes”. We have revised the manuscript based on your suggestions and made the corresponding responses according to your suggestions point by point in the following. All the changes have been revised in the manuscript.

Abstract
Point 1: Line 16: Change to "owing to".

Response 1: We sincerely thank you for your careful reading. We have changed "owning to" to "owing to".

Point 2: Line 17: Change to "common".

Response 2: Thank you for your careful reading. We have changed "commonly" to "common".

Point 3: Line 18: I think you mean "antibacterial activity".

Response 3: Thanks for your careful review work. We have changed "bacterial activity" to "antibacterial activity".

Point 4: Lines 21-22: Isn't it the C. acnes bacterium that changes morphologically rather than BSS? Please reformulate.

Response 4: Thank you for good suggestion. We have reviewed the manuscript and corrected "The morphological changes of BSS against C. acnes were observed by transmission electron microscopy (TEM)" to "the morphological changes in C. acnes by BSS were observed using transmission electron microscopy (TEM)".

Point 5: Lines 26 and 28: Change "BBS" to "BSS" if applicable.

Response 5: Thank you for your careful review which was helpful in refining the manuscript. We have changed "BBS" to "BSS".

Introduction

Point 1: Lines 45 to 46: Change sentence to "Its dry tubers have been commonly used in traditional medicine for centuries to treat various diseases, including lung disorders, gastrointestinal disorders, traumatic bleeding, burns, and ulcers."

Response 1: Thank you for your helpful suggestion. We have changed the sentence in the manuscript to "Its dry tubers have been commonly used in traditional medicine for centuries to treat various diseases, including lung disorders, gastrointestinal disorders, traumatic bleeding, burns, and ulcers."

Point 2: Line 49: Omit either "There’s evidence that" or "are known to".

Response 2: Thank you for good suggestion. We have removed "are known to" from the manuscript.

Point 3: Line 51: Change to "BSS has medical benefits that may be used for promising healthcare products".

Response 3: Thank you for your valuable suggestion. We have changed the sentence in the manuscript to "BSS has medical benefits that may be used for promising healthcare products".

Point 4: Line 54: Change to "The inhibitory effect …"

Response 4: Thanks for your helpful suggestion. We have changed "Then its inhibitory effect" to "The inhibitory effect".

Point 5: Lines 55 to 56: Change sentence to "This study provides new insights into how BSS may act against C. acnes."

Response 5: Thank you for your helpful suggestion. We have changed the sentence in the manuscript to "This study provides new insights into how BSS may act against C. acnes."

Materials and methods

Point 1: Line 73: Change to "polyamide chromatography column".

Response 1: Thank you for your careful review. We have changed "polyamide column chromatography" to "polyamide chromatography column".

Point 2: Line 105: Spell out numbers at the beginning of sentences (in this case change to "One hundred microlitres".

Response 2: Thanks for your valuable suggestion. We have changed "100 μL" to "One hundred microlitres" at the beginning of the sentences.

Point 3: Lines 118-119: Change to "This was followed by progressive dehydration using acetone solution", if applicable.

Response 3: Thanks for good suggestion. We have changed the sentence in the manuscript to "This was followed by progressive dehydration using acetone solution".

Discussion

Point 1: Lines 241-242: Change "extracts from natural plants" to "plant extracts".

Response 1: Thank you for your valuable suggestion. We have changed "extracts from natural plants" to "plant extracts".

Point 2: Lines 245-246: Change "for" to "because of".

Response 2: Thanks for your careful review work. We have changed "for" to "because of".

Point 3: Lines 246-247: Since there is more than one BSS, use the plural tense.

Response 3: Thank you for your helpful suggestion, we have corrected the tense to the plural tense at BSS correspondence in the manuscript.

Point 4: Lines 255-246: Change to "Using TEM, we observed significant morphological changes in C. acnes when exposed to BSS at MIC concentration...".

Response 4: Thanks for good suggestion. We have changed the sentence in the manuscript to "Using TEM, we observed significant morphological changes in C. acnes when exposed to BSS at MIC concentration...".

Point 5: Line 264: Change to "In this present study ..."

Response 5: Thank you for your careful review. We have changed …"In present study" to "In this present study ..."

We hope you'll accept these revisions. If you have any questions, please feel free to contact us. Thank you for your kind consideration of our manuscript.